# Long Term Benefits on Social Behaviour after Early Life Socialization of Piglets

**DOI:** 10.3390/ani8110192

**Published:** 2018-10-29

**Authors:** Irene Camerlink, Marianne Farish, Rick B. D’Eath, Gareth Arnott, Simon P. Turner

**Affiliations:** 1Animal Behaviour & Welfare, Animal and Veterinary Sciences Research Group, Scotland’s Rural College (SRUC), West Mains Rd., Edinburgh EH9 3JG, UK; Marianne.Farish@sruc.ac.uk (M.F.); Rick.DEath@sruc.ac.uk (R.B.D.); Simon.Turner@sruc.ac.uk (S.P.T.); 2Institute of Animal Husbandry and Animal Welfare, Department of Farm Animals and Veterinary Public Health, University for Veterinary Medicine Vienna, Veterinärplatz 1, Vienna A-1210, Austria; 3Institute for Global Food Security, School of Biological Sciences, Queen’s University, Belfast BT9 7BL, UK; g.arnott@qub.ac.uk

**Keywords:** ontogeny, social behaviour, pig, socialization, animal welfare, early life, aggression

## Abstract

**Simple Summary:**

Pig welfare is a societal concern, partly due to the intensive rearing conditions. One welfare concern is aggression between newly regrouped pigs. Aggression at weaning is reduced by putting several groups of piglets together when young; termed socialization. Information is missing about how socialization affects sow health and if the behavioural changes in piglets are long-lasting. We aimed to address these questions by studying sow udder quality and pig behaviour and growth in socialized and control groups. Pigs were socialized by either joining two litters (32 sows; 16 groups) at 14 days of age or not joining them (33 sows). At weaning, the sows of socialized groups had more udder damage than the controls. Socialized piglets had double the amount of bite injuries (skin lesions) than controls the day after socialization, but had 19% fewer skin lesions at regrouping at eight weeks old when injuries are more numerous and severe. At 11 weeks old, there was no difference between the groups. In a test for aggressiveness, socialized pigs attacked more often and quicker, showing greater confidence in agonistic skills. Socialization means additional work for farmers and may cause more udder damage, but has beneficial effects for pig behaviour and welfare at later regrouping.

**Abstract:**

Early life socialization of piglets has been shown to reduce piglet aggression at weaning, but information on sow health and long-term benefits is lacking. We aimed to assess how socialization impacts sow udder quality and long-term pig behaviour and growth. At two weeks of age, 65 litters either experienced socialization with one other litter (SOC) or did not (control; CON). Sows (housed in farrowing crates) were scored for teat damage and piglets were observed for aggressive behaviour (resident-intruder test) and growth and skin lesions up to 11 weeks under conventional farm conditions (including weaning and regrouping). At weaning, SOC sows had more teat damage than CON sows (*p* = 0.04). SOC piglets had double the number of lesions 24 h post-socialization compared to the control (19 versus 8; *p* < 0.001). In the resident-intruder test, more SOC pigs attacked the intruder (SOC 78%; CON 66%; *p* < 0.01), and attacked more quickly (*p* = 0.01). During regrouping (week 8), SOC pigs had 19% fewer lesions (SOC 68; CON 84; *p* < 0.05), but three weeks later, groups did not differ. Growth was unaffected by treatment. Overall, socialized piglets seem to be equipped with greater confidence or agonistic skills, leading to fewer injuries from fighting up to at least six weeks after socialization.

## 1. Introduction

Early life socialization, also called co-mingling, has been researched from the late nineties onwards as a strategy to increase social skills or reduce aggression between piglets [1,2,3,4]. The process involves removing the barriers between the farrowing crates of two or more litters so that the piglets can freely co-mingle whilst the sows remain in their crates. Socialization is a natural phenomenon that occurs in nature around the second week of life when the piglets start to leave the nest [5]. It also occurs naturally in most outdoor systems [6] and in group lactation (i.e., multi-suckling) systems [7]. 

Aggression between pigs has been a long-standing animal welfare issue for the pig sector [8], and socialization of piglets has, with mixed results, been applied in practice on a minority of farms to alleviate the situation [9,10]. Farmers have expressed concerns about the effects of socialization on sow and piglet management, behaviour, and growth, and mentioned that they would only apply socialization if it would improve productivity [9]. To increase the uptake of this method in practice, these concerns need to be addressed.

Socialization has generally yielded better results under research conditions than under commercial conditions. For example, farmers report concerns about cross-suckling of piglets between sows [9], but the majority of the experimental studies only show a limited occurrence of this [1,2,3,11]. Research under commercial conditions, as in [4], would be valuable in overcoming this gap between research outcomes and practice. This includes an assessment of the effect of socialization on productivity. Most of the research on the costs and benefits of socialization has shown a reduction in aggression at weaning [3,12,13]. Only D’Eath studied the effect of socialization on aggressive behaviour of pigs up to seven [11] and sixteen weeks of age [14]. The longer term effect of socialization on productivity is largely unknown and to our knowledge, no studies have looked at its impact on socially stable groups. If early life socialization increases social skills, as has been suggested [1,15,16], then this may translate into improved social behaviour throughout life.

The aim of the current study was to assess the impact of early life socialization on sow and piglet productivity and on piglet social behaviour in the long term. This was studied by observing 65 sows and their litters at a research farm that operates as a commercial unit, with the pigs being followed up to 11 weeks of age.

## 2. Materials and Methods 

### 2.1. Ethical Note

This study was approved by SRUC’s animal experiments committee (ED AE 46-2016) and was carried out under a UK Home Office license (project license PPL60/4330) and in constant collaboration with SRUC’s named veterinary surgeon. The study was carried out in accordance with the recommendation in the European Guidelines for the accommodation and care of animals and UK Government DEFRA animal welfare codes, and adhered to the ASAB/ABS guidelines.

### 2.2. Animals and Housing (Lactation Phase)

Research took place at the SRUC research pig farm over six batches from January–October 2016. In total, 65 gestating sows (Large White × Landrace, inseminated with American Hampshire boar semen, PIC) were moved, five days before expected parturition, into solid floor farrowing pens. Pens measured 3.15 × 1.50 m in total, with a 2.25 × 0.55 m sow crate in the middle of the pen with a small slatted dunging area at the rear and a 0.65 × 1.50 m heated creep area for piglets at the front. A handful of fresh straw and wood shavings were provided daily as bedding and nesting material. Sows were fed a standard pelleted lactation diet incrementally twice a day over the period of parturition to weaning at four weeks, and sows and piglets had unlimited access to water. Piglets were weighed and ear tagged within 6–24 h after birth. Fostering was only applied when essential for the piglets’ survival. Piglets received a 1 mL intramuscular iron injection (Ferroferon 200 mg/mL) at day 3 of age. Tails and teeth were kept intact and males were not castrated. Piglets received creep feed from ~21 days of age. No artificial milk was provided. Over the whole study, 762 piglets were born alive to 65 sows (parity 1–9), and 79 live born piglets died or were euthanized (11.5% live born mortality) prior to day 14 and the start of the experimental phase for reasons of low viability (35%), starvation (25%), laid on by the sow (13%), lameness (10%), or other reasons (17%).

### 2.3. Socialization

Sows with their litter were equally allocated to a treatment group (32 to the socialization treatment; 33 to the control group) based on their parturition date. Only neighbouring sows with <48 h between their farrowing times, and who were healthy and provided sufficient milk as judged by adequate piglet growth, were eligible for the socialization treatment. The sufficiency of milk supply of the sow was taken into account to avoid an imbalance in milk availability between litters and piglets of two litters drinking from one sow only. Birth weight was balanced between the treatments to avoid a priori differences in health and weight between the treatment groups. Socialization took place when piglets were 14 days of age by removing the barrier between two adjacent farrowing pens. The barrier was then replaced by a similar barrier with a 0.36 × 0.75 m opening to allow piglets to move freely between the two pens (Figure 1). The opening was positioned between the middle and rear of the sow, leaving a solid barrier between the heads of the neighbouring sows to avoid stress or aggression. Once in place, the new barrier with a socialization opening remained in situ until weaning. In control pens, the piglets remained in their normal farrowing pen with their own sow.

### 2.4. Sow and Piglet Measurements (Lactation Phase)

Teat damage of the sows was scored on day 13 (pre-socialization), at 24 h after socialization, and at weaning, to monitor the effect of socializing piglets on sow udder injuries. Each sow was given a score from 0 to 3 for each teat to indicate the amount of damage (none, superficial, moderate, severe) and a score for non-functional teats (0–1) and mastitis (0–1) (details can be found in Table 1). Piglets were identified by marking a number on their back with a Pentel Jumbo Marker pen every two days from day 13 onward. Skin lesions were counted on all piglets before (day 13) and at 24 h after socialization, in accordance with [17]. The actual number of skin lesions on the front, middle, and rear of the body was counted, with a distinction between the left and right side. The total number of lesions was used here for analysis. An additional score was given for the severity of facial wounds as these are common following competition for teat access (0: no injuries; 1: small scratches; 2: injuries with scab formation; 3: large patches of merged lesions) and knee damage (0: absent; 1: present). All scores, for both sows and piglets, were obtained by a single observer.

### 2.5. Animals and Housing (Post Weaning)

Piglets (n = 683) were weaned at ~26 days of age (~4 weeks) and moved to the experimental rooms on site. From weaning until week 8, pigs were kept in their original sibling groups in pens measuring 1.90 × 5.80 m (1.0–1.1 m^2^/pig). Pens had a solid floor covered with ~5 kg of long straw and were cleaned daily and refreshed with ~3.5 kg of straw as required. Pigs had ad libitum access to water and pelleted commercial feed suitable for their age and stage of growth. Pigs were gradually, over the course of two weeks, habituated to the weigh crate and to being on their own in a pen for up to 5 min, to reduce the possibility of fear responses during subsequent weighing and resident-intruder tests.

### 2.6. Resident-Intruder Tests

At seven weeks of age, 380 pigs were tested twice as residents in a resident-intruder test (RI test) [18,19]. In an RI test, an inferior and unfamiliar opponent enters the home range of the resident animal. The latency until the resident attacks the intruder is taken as a measure of aggressiveness as a personality trait [19]. Socialization is hypothesized to increase social skills and this may result in an alteration in the agonistic strategy, which was assessed here through the RI test. The test is described in detail in [20] and is here explained briefly. Resident pigs that were individually in a separate part of their home pen were confronted with a ~23% smaller intruder pig from an unfamiliar litter. Intruder pigs were not tested as residents and were used at a maximum of twice as an intruder. The time until (1) contact and (2) attack (a bite that made contact) was noted. The test was ended (a) when an attack had occurred, either by the resident or intruder; (b) five minutes after contact without an attack, i.e., time-out; (c) when the resident or intruder showed signs of distress, such as escape attempts or high pitched vocalizations; or (d) in the case a resident or intruder was mounted five times. Mounting behaviour was recorded when one of the pigs jumped with both front legs on the other animal. As this can be stressful and injurious for the recipient, the test was immediately terminated when a pig mounted the other for the fifth time. Each resident was tested twice (referred to as test 1 and test 2), with tests occurring on consecutive days with a different intruder. Once RI tests were complete, the intruder pigs and few resident pigs (that showed signs of distress in the test) were returned to the commercial farm sheds.

### 2.7. Regrouping

At eight weeks of age, pigs (n = 369) were regrouped into commercial straw flow grower pens on site. Pigs below 12 kg (n = 11) were excluded as they required a different diet for growth. Pens measured 1.8 × 5.3 m and had a solid floor covered with ~3 kg of long straw, and were cleaned daily were cleaned daily and refreshed with straw as required. Pigs had ad libitum access to water and pelleted commercial feed suitable for their age and stage of growth. The regrouping procedure mimics commercial conditions where pigs of ~8 weeks of age are regrouped with unfamiliar pigs to move into larger pens. Groups were composed according to a 2 × 2 design in which socialized and control pigs remained discrete, but grouped according to either homogeneous body weight (minimal weight variation) or heterogeneous body weight groups (maximum variation in weight). Groups consisted of 10–13 pigs (average 12), both males and females, from three or four litters, making sure each pig had at least one sibling. Group composition was balanced between treatments. In total, there were 32 groups (18 SOC; 14 CON). Skin lesions were counted in the morning before regrouping, at 24 h after, and at three weeks after (11 weeks of age). SOC pigs were not regrouped with animals they had previously been socialized with before weaning.

### 2.8. Statistical Analyses

Data were analysed with SAS version 9.3 (SAS Institute Inc., Cary, NC, USA). Results are presented as LSmeans with standard error.

Body weight at each recording was analysed separately as a response variable in a mixed model (MIXED procedure) with treatment (SOC/CON) and sex (M/F) as fixed variables and the batch (six levels) and pen (litter or group, depending on the age) as random variables. The residuals of body weight, at each recording, were normally distributed.

Sow teat damage was averaged for the 0–3 score per teat pair (score L + R/2) and then averaged for the complete udder (average of all teat pairs) for day 14, day 15, and day 26 separately. The three resulting scores were analysed as response variables in a mixed model with treatment (SOC/CON) and parity as explanatory variables and batch as a random variable. Residuals were normally distributed.

The number of skin lesions before regrouping (including socialization) was subtracted from the number of skin lesions 24 h post-regrouping to account for skin lesions already present on the body which were not related to the group mixing. Change in skin lesion numbers was analysed as the dependent variable in mixed models. Pre-weaning lesions had treatment (SOC/CON) as a fixed variable and batch and sow (litter) as random variables. Skin lesions at regrouping (wk 8) and the steady stage (wk 11) had treatment (SOC/CON), pen weight homogeneity (homogeneous/heterogeneous weight), sex (M/F), and body weight at the day of lesion count as fixed variables, and pen and batch as random variables. The interaction between treatment and pen weight homogeneity was tested but omitted from the model as it was non-significant and reduced the model fit. The residuals of skin lesions data (count data) at socialization (d13 and d14) and the steady stage (wk 11) were skewed and were square root transformed to reach normality. The residuals of the number of skin lesions at regrouping followed a normal distribution.

Animals were tested twice in the RI tests on consecutive days, and ‘test 1’ refers to their first test and ‘test 2’ to their second test. Differences between test1 and test2 for the latency to contact and attack were tested in a paired *t*-test. Differences in the attack rate (binary) were assessed with logistic regression (Chi-Square) with attack (yes/no) as the response variable and treatment as the fixed effect. The effect of the treatment on the latency to contact was tested in a mixed model (MIXED procedure) with latency to contact as the response variable, treatment and sex as fixed variables, and batch and pen as random variables. The latency to attack was analysed using a data set which only included the animals that did attack. Then, the same mixed model was used as for latency to contact, but with latency to contact added as a covariate. For each of these models, the latency to contact and attack the intruder was log transformed to obtain a normal distribution of residuals. The reason for terminating the test was assessed in a Generalized Linear Mixed Model (GLIMMIX procedure) with multinomial distribution and a cumulative logit link function. Reason for termination (four levels: attack, time-out, distress, mounting) was the response variable and treatment and sex were the fixed variables, with batch and pen as random variables.

## 3. Results

### 3.1. Sows

Sows had a mean (± SE) of 12.7 ± 0.5 live-born piglets of 1.6 ± 0.04 kg. Litter size at weaning was an average of 10.7 ± 0.2 piglets. The treatment group (SOC) and control group (CON) did not differ from each other in these parameters (all *p* > 0.05). The average teat damage (score 0–3) increased over time, with more teat damage nearer to weaning (Figure 2). At day 14 (before socialization) and at day 15, the SOC and CON group did not differ from each other (Figure 2; *p* > 0.10). At weaning, the sows from the socialized group had a higher average score for their udder, meaning more teat damage (Figure 2; F_1,56_ = 4.53; *p* = 0.04). Twenty-three sows, 16 SOC and seven CON, had at least one teat with score 3 ‘severe damage’ (Table 2). Parity influenced teat damage at weaning (b = −0.04 ± 0.02 score; F_1,56_ = 4.08; *p* = 0.05), with older sows showing, on average, a slightly lower teat damage score. However, teat damage and the number of non-functional teats widely varied between parities, with mostly the first and fifth parity affected (Table 2). Two sows were treated for mastitis; one from the socialized group and one from the control group.

### 3.2. Piglets Pre-Weaning

No piglets suffered ill health or died from the socialization treatment. After the start of the experimental phase, but before weaning, four CON piglets and two SOC piglets had to be euthanized (average age 24 ± 3 d) for low viability (n = 4), lameness (1), and being laid on by the sow (1). Body weight did not differ between the treatment groups pre-weaning (Table 3). The number of skin lesions due to aggression before socialization (day 13) did not differ between the treatment and control groups (Figure 3; SOC 1.8 ± 0.19 sqrt; CON 1.7 ± 0.19 sqrt; F_1,637_ = 0.17; *p* = 0.68). After socialization (day 14), SOC piglets had a greater increase in skin lesions than the control piglets (Figure 3; SOC 3.8 ± 0.31 sqrt; CON 2.3 ± 0.31 sqrt; F_1,629_ = 36.74; *p* < 0.001). There were no differences between the groups in the number of piglets with face or knee damage (all *p* >0.10). On day 13, 86.4% of all piglets (both treatment groups) had no face damage, 10.1% had small scratches, 3% had face injuries with scab formation, and 0.5% had large patches of merged lesions. On day 15, after the day of socialization, 81% had no damage, 16% had small scratches, and 3% had scab formation. Knee damage was present on 43% of the piglets on day 13, whereas on day 15, this was 26%.

### 3.3. Pigs Post-Weaning

SOC piglets tended to be heavier than CON piglets in week 6 only (Table 2). Males were heavier than females up until six weeks of age (Table 2). At 24h after regrouping (week 8), socialized pigs had significantly fewer skin lesions than the control group (Figure 3; SOC 67 ± 8 lesions; CON 83 ± 8 lesions; F_1,335_ = 4.03; *p* = 0.045), corresponding numerically to 19% fewer skin lesions. There was no significant difference between the SOC and CON treatments for the number of skin lesions on the front part of the body (SOC 27 ± 4 lesions; CON 33 ± 4 lesions; F_1,335_ = 2.07; *p* = 0.15), but SOC pigs had fewer lesions in the middle of the body (SOC 24 ± 3 lesions; CON 30 ± 4 lesions; F_1,335_ = 3.97; *p* = 0.047) and on the rear (SOC 16 ± 2 lesions; CON 20 ± 2 lesions; F_1,335_ = 4.77; *p* = 0.03). The number of skin lesions on the rear, which is associated with the receipt of bullying, was greater in females than in males (males 16 ± 2 lesions; females 20 ± 2 lesions; F_1,335_ = 7.50; *p* < 0.01). There was no influence of sex, body weight, or homogeneity of body weight in the pen on the total number of skin lesions at regrouping (*p* > 0.10). Three weeks after regrouping (week 11), the skin lesion difference between the groups disappeared (Figure 3; SOC 4.08 ± 0.25 sqrt; CON 4.13 ± 0.25 sqrt; F_1,335_ = 0.06; *p* = 0.81). There was no effect of sex, body weight, or group weight homogeneity on the number of skin lesions at week 11 (all *p* > 0.10). There was no interaction between pre-weaning treatment and group homogeneity on the number of skin lesions 24 h after regrouping (F_1,335_ = 0.55; *p* = 0.46) or three weeks after (F_1,335_ = 0.62; *p* = 0.43).

### 3.4. Resident-Intruder Test

The latency for the resident to contact the intruder was shorter on the second test day compared to the first test (test 1, 24 ± 1 s; test 2, 11 ± 0.6 s; t_379_ = 10.37; *p* < 0.001), and tended to be longer in test 1 when pigs were socialized (Table 4). Among resident pigs which attacked, the latency for the resident to attack (bite) the intruder was also shorter in the second test (test 1, 111 ± 4 s; test 2, 74 ± 3 s; t_203_ = 11.20; *p* < 0.001), and was significantly shorter for socialized pigs in test 1 but not test 2 (Table 4). The latency for the resident to attack the intruder in test 1 was moderately positively correlated with the latency of test 2 (r_p_ = 0.43; *p* < 0.001). Females attacked on average 13 seconds earlier than the males in test 2 (Males 4.18 ± 0.09 log (80 s); Females 3.93 ± 0.09 log (67 s); F_1,216_ = 9.34; *p* = 0.003). In the first test, the resident attacked the intruder on 64% of occasions (SOC 66%; CON 61%; χ^2^ = 1.15; *p* = 0.28). In the second test, the resident attacked the intruder on 73% of occasions. Most notably, socialized residents attacked more frequently than control residents (SOC 78%; CON 66%; χ^2^ = 7.00; *p* = 0.008). In 31% of the tests (test 1 + test 2), the intruder attacked the resident (with an SOC resident, this was 28%, whereas with a CON resident, this was 35%).

The reason for termination of the RI test was unaffected by whether pigs were socialized or not (test 1 F_1,317_; *p* = 0.42; test 2 F_1,317_; *p* = 0.90). Tests were mostly ended by one pig attacking the other (79% in test 1; 84% in test 2), or otherwise by a time out at 5 min without contact (11%; 5%), by repeated mounting behaviour (8%; 8%), or because of a fear response (2%; 2%). This was influenced by the sex of the pig, with males showing a 3.8 fold more frequent termination due to mounting in test 1 than females and a 23 fold greater termination due to mounting in test 2 (test 1 F_1,317_; *p* = 0.06; test 2 F_1,317_; *p* = 0.008; Figure 4). The resident mounted the intruder at least once in 21% of all tests (SOC 21%; CON 22%), and this equally occurred for test 1 (21%) and test 2 (22%). In 10% of all tests, the intruder mounted the resident (SOC 10%; CON 10%). When animals mounted, they were likely to continue to do so, with 80% of the mounting animals repeating their behaviour. However, females rarely mounted and if so, they did not continue as frequently as males (Figure 4).

## 4. Discussion

Socializing piglets in early life by allowing two litters to co-mingle resulted in a small increase in aggression at the time of introduction, but resulted in an increased likelihood of a resident-intruder test attack, and (among attackers) a more rapid attack when encountering an unfamiliar pig at seven weeks of age and in substantially fewer skin lesions when regrouped a week later with unfamiliar pigs. At eleven weeks of age, when in a stable social group, there was no difference in the number of skin lesions between socialized and control pigs. Sows of socialized litters had increased teat damage after socialization and this may be a concern for farms with large litter sizes.

### 4.1. Implications for the Sow

Sows of socialized litters had more teat damage on the day of weaning than control sows, with 16 SOC sows having at least one teat with severe damage. Injuries on the teats due to biting of the piglets for milk may impair the teats to the extent that they become non-functional. This is disadvantageous for the welfare and productivity of the sow and may cause pain and affect her lifetime productivity as she can nurse fewer piglets. Older sows had slightly less udder damage than young sows, but overall teat damage and the number of non-functional teats widely varied between parities, which may be a consequence of farm management (e.g., gilt selection criteria over time). In the current study, the litter size was small to average and teat damage may be more problematic on farms with large litter sizes [21]. Sows in this study were kept in farrowing crates. Although the use of free farrowing and temporary crating pens is growing, which is associated with less teat damage [22], a comparison between the different systems when socializing piglets has not yet been made. Teat and udder damage were previously found to be less frequent in group-housed (multi-suckling) sows compared to crated sows (without applying socialization of piglets), but with atrophy of the mammary glands only occurring in the group-housed sows and not in the crated sows [23].

Beneficial effects for the piglets, in terms of less stress and injuries, could last a maximum of approximately six months, until the pigs go to slaughter. The sows, however, will remain on the farm for several years and the potential trade-offs (e.g., costs in terms of udder quality) on their behaviour and health should be considered while taking into account the long-term effect of piglet socialization [24]. On the other hand, if the replacement gilts are socialized themselves, then this may reduce aggression between the sows when they are regrouped to go into the gestation groups. Socializing piglets at a later age to avoid the impact on the sow is not recommended. Although it leads to a similar reduction in aggression at regrouping (e.g., at five months age [16]), the ‘socialization’ procedure coincides with intense aggression with a higher impact on the animals’ welfare. In such a scenario, it is likely that the costs do not outweigh the benefits and animals are unnecessarily exposed to stress and the risk of injury.

### 4.2. Benefits for Growing Pigs

Early life socialization is mainly applied as a method to reduce aggression. Aggression between pigs occurs when unfamiliar animals meet and this has been a long-standing animal welfare issue for the pig sector (reviewed by [8]). Early life socialization has been shown across several studies to reduce aggression at weaning (e.g., [3,12,13]), but evidence for a reduction of aggression at a later age, when aggression is most intense, is sparse [11,14]. Our results are in line with the similar study by D’Eath [11], with an increase in the number and speed of RI attacks, but a reduction in the number of skin lesions at regrouping. The reduction in the number of skin lesions was due to fewer lesions on the middle and rear part of the body. These body areas are predominantly associated with the receipt of unilateral bullying [17] and therefore socialized pigs showed less of this damaging behaviour. Their aggressive interactions were thus predominantly reciprocal, and this type of aggression has been shown to be beneficial for long-term social stability [25].

Socialization has been suggested to increase social skills [1,15,16], which would extend to other behaviours beyond just aggression. For example, a recent study on a large-scale commercial farm found a three-fold increase in play behaviour in socialized piglets compared to the control group [4]. Evidence for improved social skills beyond an agonistic encounter is limited and deserves further research attention. In particular, the (social) cognitive abilities of socialized pigs have not yet been reported and this will be the focus of our forthcoming work.

Socialization did not affect growth performance. That body weight is unaffected by socialization is in line with the majority of previous studies.

### 4.3. Agonistic Strategy

In the resident-intruder test, the socialized residents tended to take more time to make the first contact with the intruder (such as nose contact), but then attacked significantly more quickly in test 1. A longer period of opponent assessment followed by a faster attack has also been reported in pigs showing increased assessment abilities after gaining experience in fighting [26]. It also supports the finding of [11] that socialized pigs attack sooner than control pigs in this test. A shorter attack latency in the RI test is commonly interpreted as evidence of a more aggressive personality, as it correlates with the expression of aggression in other contexts (rodents: [27]; pigs: [28]). Although socialized pigs show a shorter attack latency, it does not lead to more aggression later on. On the contrary, they show less aggression. D’Eath [11] indeed showed that socialized pigs were more direct in resolving aggressive interactions (shorter fights, less bullying, less lesions at day 10 after mixing). The behaviour during the resident-intruder test does therefore not necessarily (only) describe a trait of aggression (e.g., [28]), but also gives information on assessment abilities [20] and social strategies [29].

Females attacked sooner in the RI test, as was also shown by [28]. Males, however, clearly had a different strategy or motivation, as they showed 14 times more mounting behaviour than females and 8% of the tests had to be terminated due to repeated mounting behaviour. Some pigs show consistently more mounting behaviour than others [30], and mounting is mostly shown by males [31] even though studies have failed to find a relationship between mounting behaviour and sex hormones [31,32]. In other studies, sexual mounting occurred most frequently and typically lasted longer and coincided more often with a scream from the recipient than other forms of mounting [29]. The type of mounting in the RI test was more likely to be related to dominance, as the mounts in the test were shorter in duration than those reported by [31]. To date, the relationship between mounting and dominance rank has not been shown (no relationship was found in [31]).

## 5. Conclusions and Recommendations

Although pig farmers have reported finding early life socialization a useful technique for reducing aggression (US farmers: [10]), they have also raised multiple concerns regarding the welfare of the sow and piglet [9]. Here, we show that early life socialization of piglets on a commercially run research farm results in increased teat damage of the sows, but a subsequent reduction in aggressive behaviour of the pigs and no constraints on productivity. Applying this strategy should, however, be done while taking into account the circumstances, such as sow and litter health, udder condition, litter size, and differences in age between neighbouring litters. If such characteristics are ignored and barriers are removed as routine practice, then this may indeed lead to an increase in, for example, cross suckling or reduced health. As research into animal behaviour is increasingly focusing on positive animal welfare, it would be worthwhile to investigate whether other behavioural and physiological changes, such as play behaviour, social cognition, and stress resilience, are altered and whether changes are temporary or permanent.

## Figures and Tables

**Figure 1 animals-08-00192-f001:**
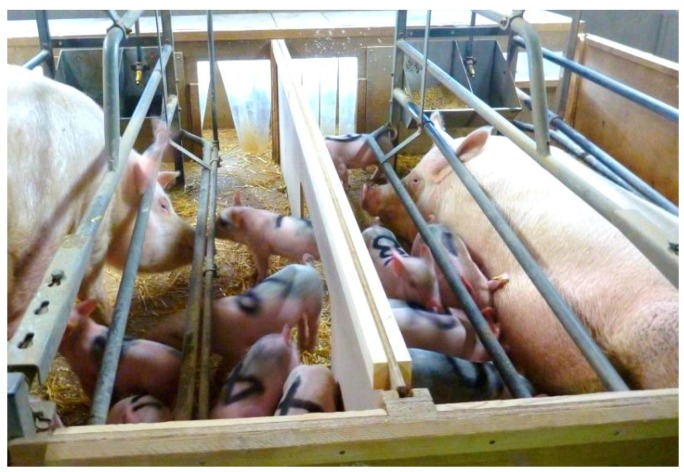
Photo taken just after the solid barrier was replaced by a wooden barrier that allowed the piglets to move freely between the two pens. Photo credit: M. Farish.

**Figure 2 animals-08-00192-f002:**
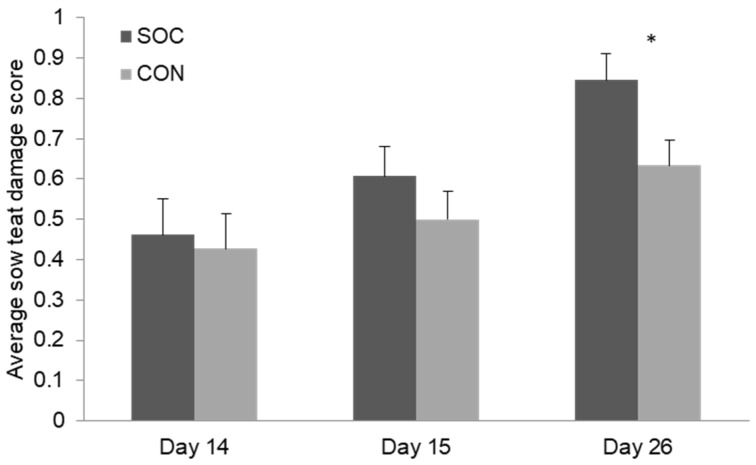
Average score for teat damage of the sow by treatment group (socialized: SOC; control: CON) before socialization (D14), 24h after socialization (D15), and at weaning (D26). n = 64. * Means differ by *p* < 0.05.

**Figure 3 animals-08-00192-f003:**
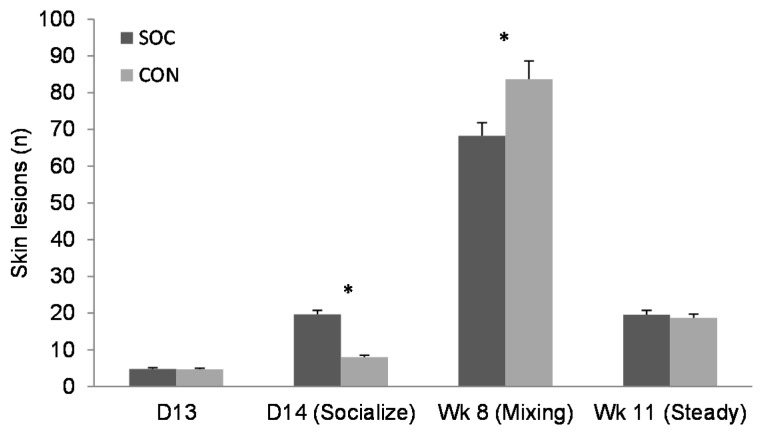
Untransformed means with SE for the number of skin lesions for the socialized group (SOC) and control group (CON). LS means with *p*-values are given in the text. * Means differ by *p* < 0.05.

**Figure 4 animals-08-00192-f004:**
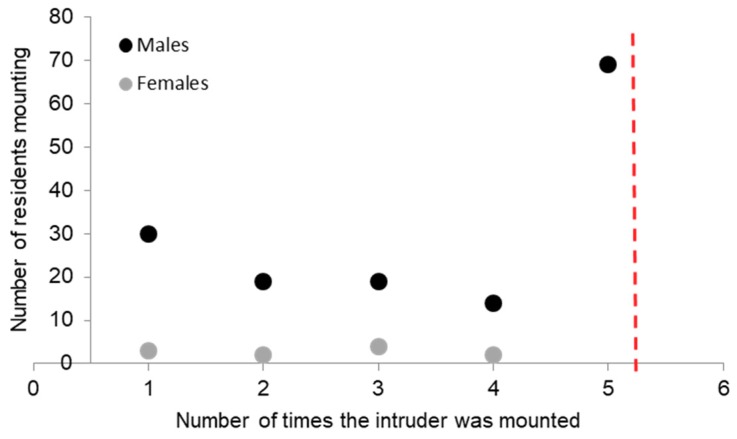
Frequency of mounting behaviour during tests by resident males (black) and females (grey). The test was immediately terminated after the *fifth mount (red dashed line).*

**Table 1 animals-08-00192-t001:** Teat damage score.

Score	State	Description
0	Normal	Normal healthy teat with potential to give milk, but which may be unused at this time (dry). Surrounding tissue is clean.
1	Superficial	Teat has small lesion or minor redness but without swelling or infection. Minor lesions may exist on the surround tissue.
2	Moderate	Teat is damaged by a medium or deep lesion or injury with broken skin which may or may not be swollen. Lesions or bite marks could also be present on surrounding tissue.
3	Severe	Teat is severely injured and may or may not be bleeding. Inflammation or infection may be present. Lesions or bite marks could also be present on surround tissue.
NF	Non-functional	Teat is inverted or damaged due to necrosis or is malformed. Teat is not providing milk.
M	Mastitis	Infection/inflammation on udder causing normally healthy udder to be swollen and providing less milk. May in addition have scores 0–3.

**Table 2 animals-08-00192-t002:** Number of sows (/out of total number per group) with at least one teat with score 3 at weaning or at least one non-functional teat at weaning, for sows in the socialized and control group by parity. Max. teats gives the maximum number of affected teats per sow by parity.

	Score 3 Teat Damage			Non-Functional Teats		
Parity	Socialized	Control	Max. Teats	Socialized	Control	Max. Teats
1	4/5	2/3	4	3/5	3/3	2
2	1/3	1/3	4	2/3	2/3	4
3	5/6		2	3/6		2
4	1/2	2/3	3	1/2	1/3	2
5	5/6	1/4	3	4/6	4/4	2
6		1/2	1		2/2	1

**Table 3 animals-08-00192-t003:** LSMeans with SE and *p*-value for body weight of socialized and control piglets and for males and females.

Weight	n	Socialized	Control	*p*	Males	Females	*p*
D0 *	683	1.6 ± 0.0	1.6 ± 0.0	0.31	1.7 ± 0.0	1.6 ± 0.0	0.001
W4	683	8.3 ± 0.3	8.4 ± 0.3	0.55	8.5 ± 0.2	8.2 ± 0.2	0.002
W6	683	14.7 ± 0.3	14.1 ± 0.3	0.09	14.7 ± 0.2	14.1 ± 0.2	0.002
W7	369	21.9 ± 0.4	21.7 ± 0.5	0.46	21.9 ± 0.4	21.7 ± 0.4	0.31
W11	369	44.0 ± 0.7	43.6 ± 0.8	0.50	44.0 ± 0.7	43.5 ± 0.8	0.25

* Excluding piglets that died before weaning.

**Table 4 animals-08-00192-t004:** Latency for the resident to contact and to attack the intruder, in test 1 and test 2, for pigs that were socialized (SOC) or not socialized (CON). Values are LSmeans with SE for log transformed values, whereby raw values in seconds are given in brackets.

Latency	SOC	CON	Test Statistics	*p*
Test 1: Contact latency	2.88 ± 0.06 (26 s)	2.71 ± 0.07 (21 s)	*F*_1,319_ = 2.99	0.08
Test 1: Attack latency *	4.62 ± 0.06 (103 s)	4.66 ± 0.07 (121 s)	*F*_1,183_ = 6.23	0.01
Test 2: Contact latency	1.98 ± 0.08 (11 s)	2.04 ± 0.09 (11 s)	*F*_1,318_ = 0.29	0.59
Test 2: Attack latency *	4.00 ± 0.09 (72 s)	4.11 ± 0.10 (76 s)	*F*_1,216_ = 1.19	0.28

* Without non-attackers.

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
