# Peer review of "Long Term Benefits on Social Behaviour after Early Life Socialization of Piglets"

_animals, 2018, doi:10.3390/ani8110192_

Round 1

Reviewer 1 Report

The authors investigated the effect of early life socialisation of piglets on sow udder damage as well as lesions in piglets caused by aggressive interactions during socialisation and at weaning. The study is well designed and the sample size appropriate. The methods are described in detail, and the results presented adequately. I have two comments.

Line 88 and Figure 1: The sows were kept in farrowing crates which may have impaired their ability to react to piglets fighting at the udder compared to sows kept in free farrowing systems. Crating may thus have contributed to the incidence of udder damage in the SOC treatment. In my view, this issue could be addressed in the Discussion. Are there studies on early life socialisation of piglets whose mothers were kept in loose farrowing pens?
Moreover, I suggest to mention in the Abstract that the sows were housed in farrowing crates.

Lines 176-178: When piglets were mixed at the age of 8 weeks, the newly composed groups consisted of individuals from 3 or 4 litters and contained both litter mates and unfamiliar pigs. Did the authors analyse whether the proportion of unfamiliar pigs in these groups had an effect on the incidence of skin lesions 24h postregrouping?
Are there studies showing that mixing socialised (and thus familiar) piglets at weaning reduces the incidence of skin lesions? Possibly, such selective mixing would increase the long term benefit of early life socialisation.

Author Response

Response to Point 1:

The abstract now includes that the sows were crated. Socialization naturally occurs in multi-suckling systems where the sows are loose, but multiple litters intermingle and more cross-suckling may occur. Because this is such a different system it cannot be compared directly to the situation when only two litters are put together. In the discussion is now included:  “Sows were in this study kept in farrowing crates. Although the use of free farrowing and temporary crating pens is growing, which will give a sow more freedom to move and to protect her udder, a comparison between these different systems when socializing piglets has not yet been made. Teat and udder damage were previously found to be less in group-housed (multi-suckling) sows compared to crated sows (without applying socialization of piglets), but with atrophy of the mammary glands only occurring in the group-housed sows and not in the crated sows (Hultén et al., 1995).”

Response to point 2:

The effect of proportion of unfamiliar pigs on skin lesions was not analysed as unfamiliairty was balanced between the treatment groups (now mentioned in methods). Groups with a higher degree of unfamiliarity have more fights (e.g. Arey & Franklin, 1995), but in large groups pigs initiate less fights (Turner et al., 2001).

Socializing piglets has shown to reduce aggression at weaning. The sentence in the introduction mentioning these references has been slightly reworded to clarify this. “Most of the research on the costs and benefits of socialization has shown a reduction in aggression at weaning [3,12,13]. Only D’Eath studied the effect of socialization on aggressive behaviour of pigs up to seven [11] and sixteen weeks of age [14].”  

Reviewer 2 Report

This is a timely study into the effect of early socialisation on piglet and sow productivity and welfare that I believe adds to the current knowledge on the consequences of socialisation on commercial productivity.

Some minor points to address:

Can the authors confirm how pigs were picked for regrouping? Line 149 states 380 pigs were tested as residents and line 169 states 369 were regrouped. Were all regrouped pigs residents? Or were intruders also regrouped? If regrouping included both residents and intruders did their status in the RI test affect the lesion score after regrouping?

The results state that older sows had lower teat damage, this is an interesting result yet is not discussed. Is the damage to lower parity sows such that it has the potential to reduce their productivity (either by reducing the number of litters they can support or by reducing the size of litter they can support?). Increased teat damage is clearly a welfare issue and some sort of cost/benefit analysis would have perhaps been useful.

Author Response

Response to point 1 (regrouping):

Pigs below 12 kg were excluded from regrouping as they could not yet join the heavier pigs on the same diet composition. This has now been added to the methods. Only residents continued in the trail after the resident-intruder test. Intruders were directly after the RI test returned to the commercial stock, which was mentioned in the methods (line 167).

Response to point 2 (teat damage):

Extra details on teat damage have been given in the results section (Twenty-five sows, 18 SOC and 7 CON, had at least one teat with score 3 ‘severe damage’). This has been discussed as being a potential risk factor for reduced productivity.

Extra information on teat damage and non-functional teats in relation to parity has been provided in the results. The difference in teat damage regarding parity has now been mentioned in the discussion, but with the note that udder damage and the number of non-functional teat varied widely between parities and that this may be a consequence of specific farm management (e.g. changes in gilt selection criteria over time) rather than being related to socialization.

Thank you for reviewing the manuscript.